# Targeting the Ubiquitylation and ISGylation Machinery for the Treatment of COVID-19

**DOI:** 10.3390/biom12020300

**Published:** 2022-02-12

**Authors:** George Vere, Md Rashadul Alam, Sam Farrar, Rachel Kealy, Benedikt M. Kessler, Darragh P. O’Brien, Adán Pinto-Fernández

**Affiliations:** 1Target Discovery Institute, Centre for Medicines Discovery, Nuffield Department of Medicine, University of Oxford, Roosevelt Drive, Oxford OX3 7FZ, UK; gv272@exeter.ac.uk (G.V.); md.alam@jesus.ox.ac.uk (M.R.A.); sam.farrar@bnc.ox.ac.uk (S.F.); benedikt.kessler@ndm.ox.ac.uk (B.M.K.); 2MRC Centre for Medical Mycology, University of Exeter, Geoffrey Pope Building, Stocker Road, Exeter EX4 4QD, UK; 3Environmental Futures & Big Data Impact Lab, University of Exeter, Stocker Rd., Exeter EX4 4PY, UK; rk480@exeter.ac.uk; 4Chinese Academy for Medical Sciences Oxford Institute, Nuffield Department of Medicine, University of Oxford, Roosevelt Drive, Oxford OX3 7FZ, UK

**Keywords:** SARS-CoV-2, COVID-19, ubiquitin proteasome system, ISGylation, ubiquitomics

## Abstract

Ubiquitylation and ISGylation are protein post-translational modifications (PTMs) and two of the main events involved in the activation of pattern recognition receptor (PRRs) signals allowing the host defense response to viruses. As with similar viruses, SARS-CoV-2, the virus causing COVID-19, hijacks these pathways by removing ubiquitin and/or ISG15 from proteins using a protease called PLpro, but also by interacting with enzymes involved in ubiquitin/ISG15 machinery. These enable viral replication and avoidance of the host immune system. In this review, we highlight potential points of therapeutic intervention in ubiquitin/ISG15 pathways involved in key host–pathogen interactions, such as PLpro, USP18, TRIM25, CYLD, A20, and others that could be targeted for the treatment of COVID-19, and which may prove effective in combatting current and future vaccine-resistant variants of the disease.

## 1. Introduction

Coronaviruses (CoVs) are enveloped, single-stranded, positive-sense RNA viruses that cause the common cold in a broad range of mammals and avians [1]. Severe infection can lead to respiratory and multi-organ failure, as well as digestive and neurological insults [2]. This was previously evidenced with the emergence of the pathogens responsible for Severe Acute Respiratory Syndrome coronavirus (SARS-CoV) in 2002 [3], and Middle East respiratory syndrome coronavirus (MERS-CoV) in 2012 [4]. Both diseases were linked to zoonotic origins and resulted in a highly contagious and sometimes lethal acute respiratory illness. In December 2019, the World Health Organization (WHO) was alerted to patient clusters of a pneumonia of unknown cause in the city of Wuhan, Hubei Province, China [2]. A novel coronavirus species, SARS-Cov-2, was subsequently isolated and identified, resulting in a highly contagious condition now commonly known as COVID-19 (Coronaviridae Study Group of the International Committee on Taxonomy of Viruses, 2020; [5]). The exact origins of SARS-Cov-2 are still unclear, but it is suspected to have ancestral origins in bats [1]. As of January 2022, it is estimated that ~300 million cases have been detected globally, which has resulted in ~6 million deaths (www.google/globalCOVIDcases; accessed on 28 January 2022). 

COVID-19 has seen an unprecedented level of research and funding dedicated to fighting the disease. Novel viral vector and nucleic acid vaccines have been developed in record times, expedited by knowledge harbored from preceding outbreaks. Despite the success of global vaccine rollouts, several mutated variants of SARS-CoV-2 have evolved, resulting in subtle changes in disease indications and strengths of transmissibility [6]. Due to the likely emergence of future vaccine-resistant variants, it is imperative to develop novel strategies to combat the disease. The post-translational modification of proteins with ubiquitin (ubiquitylation) and/or ISG15 (ISGylation) plays a key role in mediating cellular host–pathogen interactions and antiviral signaling and defense by modulating key events of the innate immune activation signaling. In this review, we discuss how the machinery involved in ubiquitylation and ISGylation can be utilized for the development of novel therapeutics against SARS-CoV-2 infection. 

## 2. SARS-CoV-2-General Biology and Mechanisms of Infection

### 2.1. SARS-CoV-2 Viral Genome Architecture 

SARS-CoV-2, along with other CoVs, has a genome of approximately 30kB of single-stranded, positive-sense RNA [7], making them the largest RNA genomes described to date [8]. The virus has 79% sequence identity with SARS-CoV and 50% with MERS-CoV [9]. Its genome contains twelve functional open reading frames ordered from 5′–3′ that encode the viral replication transcription complex (ORF1a/1b), the four structural proteins, Spike (S), Envelope (E), Membrane (M), and Nucleocapsid (N), and the likelihood of highly variable accessory proteins encoded throughout [7] (Figure 1A). Once viral RNA is released into the cell, it translates ORF 1a/1b to generate the huge replicase polyproteins pp1a and pp1ab, which are responsible for viral transcription, replication, and higher-order RNA structure [7]. Sixteen non-structural proteins (nsps) are liberated by the proteolytic cleavage of pp1a (nsp1–11) and pp1ab (nsp1–10, nsp12–16). The enzymes responsible for this are two cysteine proteases; papain-like protease (PLpro) is positioned in the large nsp3 subunit, and the highly conserved main protease (Mpro)/chymotrypsin-like protease (3CLpro) is located within nsp5 (Figure 1B). Mpro cleaves the nsp4–nsp11 region of pp1a and nsp4–nsp16 of pp1ab, whereas PLpro cleaves the nsp1–nsp4 domain [10]. 

### 2.2. SARS-CoV-2 Mode of Entry and Proliferation 

An overview of the SARS-CoV-2 viral life cycle is depicted in Figure 1A. As with the original SARS-CoV virus, human SARS-CoV-2 uses angiotensin-converting enzyme 2 (ACE2) as the main receptor for host cell invasion [11,12]. More recently, CD147 has also been identified as a novel route for infection [13]. The heavily glycosylated S protein mediates this attachment to host cell surface receptors [5]. Similar to other class I fusion glycoproteins, the viral S protein is cleaved and activated by host cell proteases to form an endosome [14]. The protease central to SARS-CoV-2 cell entry is Transmembrane Protease, Serine 2 (TMPSS2; Figure 1A) [15]. Once the endosome is formed, SARS-CoV-2 enters the cell either through acidification or the action of host protease cathepsin L. 

The S protein is composed of two functional subunits: S1, which is responsible for binding to membrane-bound ACE2 [16], and S2, which contains the fusion domain. In the C-terminal to the S1 domain lies the receptor-binding domain (RBD), which is essential for viral entry [12,15,17]. One feature of SARS-CoV-2 that sets it apart from SARS-CoV and other beta-coronaviruses is a four-residue insertion of PRRA, forming a polybasic cleavage site RRAR at the junction of S1 and S2 (Figure 1B). This cleavage site can recruit furin and other host proteases, but it is unclear how this affects general virulence [17].

### 2.3. Clinical Manifestations of COVID-19 

SARS-CoV-2 causes a wide spectrum of disease, from asymptomatic illness to severe acute respiratory failure. Findings on autopsy in terminal cases include diffuse alveolar damage, interstitial edema, and reactive type II pneumocytes [18]. Since SARS-CoV-2 enters cells via ACE2, there is a gradient of infection across the respiratory tract, with cells in the upper respiratory tract with high ACE2 and TMPRSS2 expression showing higher viral loads [19]. It is increasingly recognized that the host immune response is a major factor in the clinical manifestations of COVID-19. Indeed, aspects of the immune system are evaded, whereas others are magnified, leading to profound cytokine release, T cell activation, increased antibodies, and abnormalities of the granulocyte lineage [20]. In severe cases, T cells are both depleted and strongly activated [21], and peripheral blood has a “cytokine storm” profile, with large concentrations of inflammatory cytokines [20]. How SARS-CoV-2 achieves this in vivo has not been fully characterized, but work on airway epithelial cell lines has revealed a complex interaction of delayed, but powerful, type I interferon (IFN) response to SARS-CoV-2 infection, perhaps as a way of gaining time for viral replication [22]. Accumulating evidence has demonstrated that numerous IFN-stimulated genes (ISGs) and post-translational ubiquitylation play key roles in cellular antiviral signaling and defense by modulating key events of the innate immune activation signaling. Such mechanisms provide an attractive framework for the development of SARS-CoV-2 therapeutics. 

## 3. Ubiquitylation, ISGylation and Their Roles in Human Antiviral Responses

### 3.1. Ubiquitylation

The modification of proteins by ubiquitin and ubiquitin-like proteins (Ubls) plays a key role in mediating cellular antiviral signaling and defense. Ubiquitin is an 8.6 kDa protein, and its conjugation to lysine residues on cellular proteins controls many processes, especially targeted protein degradation by the proteasome [23,24]. There is an extensive network of enzymes that controls the addition and removal of ubiquitin to substrates, which are carried out by the approximately 600 ubiquitin E3 ligases and 100 deubiquitinases (DUBs) in humans [25]. The substrate specificity of these enzymes enables varied outcomes of ubiquitin modification. In addition to its N-terminus, ubiquitin itself has seven acceptor lysine (K) sites to which further ubiquitin proteins can be conjugated, allowing the formation of eight basic types polyubiquitin chains. The combinatorial possibilities of ubiquitin chains have led to speculation that their structure can confer signaling information in the cell [26]. Recent technologies have begun to reveal an unprecedented landscape of ubiquitin chain composition [27,28].

### 3.2. Ubiquitin in Innate Immune Sensing Pathways

Polyubiquitin chains are vital mediators in signaling downstream of three of the main families of innate immune signaling pathways, known as pattern recognition receptor (PRRs): toll-like receptor (TLRs); retinoic acid-inducible gene 1 (RIG)-like receptors (RLRs); and nucleotide-binding oligomerization domain (NOD)-like receptors (NLRs), which act together to sense a variety of Pathogen-Associated Molecular patterns (PAMPs). These pathways all require the assembly of K48- and K63-linked polyubiquitin chains on signaling proteins downstream of PRRs [29] (Figure 2). Triggering type I IFN production and release is a crucial outcome of PRR signaling, as type I IFN unleash autocrine and paracrine antiviral responses [30]. Research is currently defining the precise role of PRRs in sensing SARS-CoV-2, building on the knowledge of coronavirus infection [31]. Therefore, this contributes to the understanding of ubiquitin in antiviral signaling and defense in COVID-19 [32]. 

The balance of signaling is crucial. Although PRR signaling is key for viral sensing and clearance, an excessive inflammatory response leads to pathological lung inflammation. Proinflammatory cytokines produced downstream of TLR-driven NF-kB signaling pathways are highly elevated in severe cases of COVID-19. In the lung airway space, bronchoalveolar lavage (BAL) washes allows recovery of supernatant for cytokine readouts. These have shown increased Interleukin (IL)-6 and IL-8 production in severe cases COVID-19 [33]. Plasma IL-6 is highly elevated, but GM-CSF is suggested to be a particular feature of severe cases of COVID-19, as it not found in patients who have died from influenza virus infection [34]. Although immune cells are the most likely cellular source of these cytokines, their precise origin has yet to be elucidated. Understanding which cell types are producing these cytokines will give clues to which viral sensing pathways are hyperactivated in COVID-19.

### 3.3. TLR Signaling

TLR receptors can detect a variety of viral PAMPs [35], and current work is uncovering how SARS-CoV-2 activates TLRs. On first contact with cells, exposed proteins on the surface of SARS-CoV-2 can activate TLR2 and TLR4 (Figure 2). Ubiquitin is important in TLR signaling from surface receptors. Once TLR2 or TLR4 bind a PAMP, the adaptor protein MyD88 is recruited to the intracellular domain of the receptor, where it oligomerizes and forms the basis of a protein complex known as the Myddosome [36]. Several members of the IL-1R-associated kinase (IRAK) family are recruited to the Myddosome mediated by death domain interactions of the proteins. The Myddosome platform then recruits E3 ligases TNF receptor-associated factor 6 (TRAF6), Pellino1, and Pellino2, which assemble K63-linked polyubiquitin chains on IRAK1. TRAF6 recruits linear ubiquitin chain assembly complex (LUBAC), which assembles Met1-linked ubiquitin chains. The resulting K63-Met1 hybrid chains serve as a signaling platform that activates the TAK1/TAB2/3 complex and NEMO/IKKα complex, leading to inflammatory gene expression [37].

Although TLR4 is mostly known for detecting lipopolysaccharide from Gram negative bacteria, it can bind viral PAMPs such as respiratory syncytial virus (RSV) F protein [38], causing proinflammatory signaling. TLR4 is suggested to also sense SARS-CoV-2 S protein [39]. In response to S protein, bone-marrow-derived macrophages (BMDMs) from TLR4 knockout mice do not induce the same level of IL-1β mRNA expression as wild-type BMDMs. Further evidence will be needed to confirm this in vivo. 

TLR2 has been found to detect the envelope protein of SARS-CoV-2, and induces proinflammatory cytokine production [40]. In severe COVID-19, the transcriptome of whole blood shows more mRNA of several TLRs, including TLR2 and the TLR adaptor protein MyD88 in severe COVID-19 [40]. This signature could be due to increased numbers of circulating immune cells confounding bulk transcriptomic analysis, rather than upregulation of the mRNAs on an individual cell basis. Nevertheless, this points to an increased capacity for TLR signaling in the blood, suggesting pathological upregulation of TLR signaling in severe COVID-19. 

The binding of dsRNA to TLR3 in endosomes results in assembly of a signaling complex including recruitment of ubiquitin ligases such as Pellino1 [41] and LUBAC. Pellino1 knockout BMDMs do not upregulate IFN-β production [42], and LUBAC deficiency causes impaired TLR3 signaling during influenza virus (IAV) infection [43]. Pellino1 has been shown to assemble K63-linked polyubiquitin chains on IRAK-1 in vitro, although this is not yet proven to have role in TLR3 signaling. 

The SARS-CoV-2 replication cycle involves a dsRNA intermediate [44], which is an activator of TLR3 and TLR7 signaling. The transcription factor interferon regulator factor (IRF) 3 is activated by these pathways, and drives antiviral IFN-I gene expression. In COVID-19, rare genetic variants in TLR3 and TLR7 have been discovered in patients with severe disease, suggesting the importance of these pathways [45,46]. TLR3 knockout mice are more susceptible to SARS-CoV infection [47], and TLR3/7 knockdown impairs SARS-CoV-2 sensing of Calu-3 cells [48].

### 3.4. RLR Signaling

Coronaviruses have previously been identified as activators of RLR signaling in murine models [49,50]. RLR signaling detects viral RNA and depends on assembly of polyubiquitin chains for signaling [51]. There are three sensors in the RLR family, all of which are cytosolic: RIG-I and MDA5 initiate a signaling pathway that causes production of type I IFNs, whereas LGP2 regulates RIG-I signaling. When RIG-I is inactive, the Caspase Activation and Recruitment Domains (CARD) are sequestered in the protein. However, on binding RNA molecules with a 5′ triphosphate group, the CARD is released and mediates downstream signaling with Mitochondrial Anti-Viral-Signaling protein (MAVS). Less is known about the mechanism of MDA5 activation, with single- or double-stranded RNA proposed as a ligand. 

RIG-I activation depends on a carefully tuned network of E3 ligases and DUBs to promote signaling and balance degradation. The CARD domain opening in RIG-I depends on K63-linked polyubiquitylation. A structure of RIG-I-Ub2 (diubiquitin) has shown the mechanism of how polyubiquitylation mediates the signaling. Non-covalent interactions between the ubiquitin and CARDs enable tetramerization of the CARD subunits, generating a binding surface that can activate a CARD on MAVS, allowing downstream signaling [52]. The ubiquitylation of RIG-I is controlled by several E3 ligases, including TRIM25 and TRIM4 [53,54], with redundancy likely being important due to selective pressure from viruses. Signaling is inhibited by the DUBs CYLD and USP3, which remove the K63-linked chains [55,56]. In addition to controlling signaling, ubiquitylation also regulates protein levels through K48-linked ubiquitin chains in RLR signaling. RIG-I can be ubiquitinated with K48-linked chains by Riplet [57]. RIG-I and TRIM25 can be stabilized through removal of K48-linked chains by USP4 and USP15, promoting signaling [58,59]. 

The role of ubiquitin in MDA5 signaling is less clear. MDA5 is activated by K63-linked ubiquitylation of its helicase domain [60], and its CARDs have a lower affinity for K63-linked ubiquitin, suggesting a different mechanism than RIG-I activation. In SARS-CoV-2 infection, MDA5 activation has also been found to depend on modification with a ubiquitin-like protein (Ubl), ISG15 [61], which will be expanded in the section on Ubls in antiviral defense.

The mechanism of SARS-CoV-2 sensing by RLRs is currently being elucidated. Most studies so far have looked at RLR signaling in Calu-3 cells, a lung adenocarcinoma cell line that expresses the factors required for SARS-CoV-2 entry, TMPRSS2 and ACE2 [62,63,64,65]. These papers all show the importance of MAVS and thus RLR signaling in the detection of SARS-CoV-2. Three of these studies suggest that MDA5 is the key sensor and RIG-I is dispensable [63,64,66], but Thorne et al. suggest that RIG-I depletion does affect SARS-CoV-2 sensing [64]. A study of primary bronchial epithelial and alveolar cell infection by SARS-CoV-2 suggest the opposite, that RIG-I sensing is required whereas MDA5 is dispensable [66]. More studies will be required in different cell lines and in vivo research to clarify whether RIG-I is contributing to SARS-CoV-2 sensing. 

Once activated by RIG-I or MDA5, MAVS forms filaments on the surface of the mitochondria [51]. This complex activates the transcription factor dimer IRF3/7, which drive expression of IFNs and ISGs. MAVS is also ubiquitylated by TRIM25, however in this case, TRIM25 assembles K48-linked chains on MAVS which cause its proteasomal degradation. MAVS degradation is required to release a signaling complex including NEMO and TBK1 [67]. K63-linked ubiquitin chains on NEMO serve as a platform to activate the kinase TBK1 and allow phosphorylation of IRF3. SARS-CoV-2 can modulate NEMO ubiquitylation to oppose MAVS signaling, which will be explored in the section on the viral modulation of ubiquitin signaling [68].

### 3.5. Ubls in Antiviral Defense: ISGylation

Ubiquitin is part of a wider protein family, the Ubls. Similar to ubiquitin, Ubl modification regulates many cellular processes [69,70,71]. The most important Ubl in antiviral defense is Interferon Stimulated Gene 15 (ISG15), induced by IFN-I signaling. Its conjugation to both host and viral proteins mediates protective antiviral effects through various mechanisms [72], which are also relevant to SARS-CoV-2 infection.

ISG15 consists of two Ubl domains, and is a rapidly evolving protein. Ubiquitin is almost completely conserved between yeast and humans, differing by only 3 conservative mutations, whereas human and mouse ISG15 has only 66% sequence similarity. This could be a result of selective pressure from viruses on ISG15. Similar to ubiquitin, ISG15 is conjugated to proteins on lysine residues by four E3 ligases, EFP, TRIM25, HHARI, and HERC5 in a process called ISGylation. HERC5 is the main E3 ligase for ISGylation in humans, interestingly, and highlighting again the variability of the pathway between different organisms, mouse express HERC6 instead of HERC5 [73,74]. Of these, HERC5 is relevant in the antiviral setting. HERC5 widely modifies the proteome with ISG15, in particular targeting proteins as they are produced in ribosomes [75]. Both host and viral proteins are ISGylated. The modification of host proteins tunes the antiviral response. For example, ISGylation activates the kinase PKR, which phosphorylates eIF4α and inhibits translation [76]. ISGylation of viral proteins impairs their function, inhibiting binding activity [77] and disrupting oligomerization of viral proteins, such as the L1 capsid of human papillomavirus [78]. 

SARS-CoV-2 infection induces ISG15 expression and its conjugation apparatus in induced pluripotent cell-derived (iPSC) macrophages [79]. MDA5, but not RIG-I activation, has been found to require ISGylation on two lysine residues in its CARD domain. Preventing MDA5 ISGylation by mutating the modified lysine residues to arginine decreases ISG expression, as RLR signaling is attenuated. In HEK293 cells infected with SARS-CoV-2, prevention of MDA5 ISGylation increases viral PFUs by 9-fold [61]. However, ISGylation can be opposed by SARS-CoV-2, which we will explored in the next section. Finally, ISG15 can also act as a cytokine, acting on immune cells through LFA-1 to trigger IFN-γ release [80]. Infection of iPSC macrophages with SARS-CoV-2 causes ISG15 secretion through a novel autophagy-dependent pathway [79]. In these cells, free ISG15 promotes the release of pro-inflammatory cytokines. 

### 3.6. Viral Modulation of Ubiquitin and ISG15 Signals

Viruses need to evade innate immune responses to replicate, and achieve this through methods such as hiding PAMPs, inhibiting viral sensing pathways, shutting off host translation, and as we will consider here, modulating the ubiquitin and ISG15-mediated responses [81]. Systemic mapping of SARS-CoV-2 host–protein interactions suggest that SARS-CoV-2 proteins bind to a wide set of host proteins including MAVS binding by ORF7b, although the functional consequences of this particular interaction were not explored [82]. Other studies have explored the functional consequences of viral host–protein interactions. SARS-CoV-2 ORF9b interacts with NEMO and blocks its K63-linked ubiquitylation, directly inhibiting RLR signaling [68] (Figure 2). 

Two studies have investigated the ubiquitylome of SARS-CoV-2-infected cells. The ubiquitylome is the set of proteins modified by ubiquitin in a cell under a given set of conditions [83]. These studies show broad changes in the ubiquitylome, probably reflecting the battle between host and virus [82,84]. In addition, both of these studies detect ubiquitylation of SARS-CoV-2 proteins, although it is not clear how this affects their function. Ubiquitomic methods rely on the detection of the remnant diGlycine (GG) of the C-terminus of ubiquitin conjugated to the protein after treatment with trypsin. However, ISG15 also leaves the same GG remnant; therefore, several sites detected in these studies could be from ISG15. 

There are many examples of viral proteases that cleave ubiquitin and ISG15 from substrates [85,86,87]. As described earlier, SARS and SARS-CoV-2 both express two proteases, Mpro and PLpro. The latter can cleave both ubiquitin and ISG15 from protein substrates [88,89,90]. SARS-CoV-2 PLpro is a multifunctional protease, cleaving several viral encoded proteins nsp1, nsp2 and nsp3 to enable their maturation [88]. Biochemical assays have shown functional differences between SARS and SARS-CoV-2 PLpro. Incubation of PLpro with purified ubiquitin chains of various linkages and ISG15 shows that SARS-CoV-2 PLpro preferentially cleaves ISG15 substrates, with a small specificity for K48-linked chains and not cleaving other ubiquitin chain linkages [88]. Ubiquitin and ISG15 Activity-Based Probes (ABP) have further shown the specificity of PLpro. ABPs are purified proteins that have been chemically modified to include a warhead that reacts irreversibly with an enzyme and include a tag to enable visualization. The type of purified protein gives specificity, e.g., an ABP that is made from ISG15 will react with deISGylating enzymes. SARS-CoV-2 PLpro reacts most strongly with ISG15 ABPs, with limited reactivity again ubiquitin ABPs, again suggesting that ISG15-modified proteins are the major target of this enzyme [90]. This is in contrast to SARS PLpro, which exhibits greatest specificity for K48-linked ubiquitin chains. Structures of ISG15-bound SARS-CoV-2 PLpro have informed mutagenesis experiments, suggesting the mechanism of ISG15 specificity [90,91,92]. ISG15-conjugation is a key regulator of viral signaling; therefore, this likely represents a strategy to dampen immune responses (Figure 2).

The consequences of the ISG15-cleaving activity of SARS-CoV-2 PLpro are being explored. The study of PLpro in viral infected cells is challenging, as the proteolytic activity of PLpro is also required for cleavage of the viral precursor proteins. Therefore, studies have focused on PLpro activity in transfected cells. IRF3 is ISGylated as part of the IFN-I signaling pathway. However, the expression of SARS-CoV-2 PLpro in IFN-α-treated cells decreases IRF3 ISGylation, showing that PLpro can cleave protein-conjugated ISG15 [90]. SARS-CoV-2 opposes MDA5 ISGylation, which (as discussed) is required for RLR signaling [61]. A proteome-wide study has suggested broader targets of PLpro by comparing detected GG sites in WT and ISG15 knockout IFN-I treated macrophages when PLpro is expressed [79]. Glycolytic enzymes are found to be ISGylated which is abolished by PLpro expression. The authors suggest ISGylation of glycolytic enzymes inhibits their activity and restricts glycolytic flux, in line with a previous report on the role of ISGylation in regulating metabolism [93]. The deISGylating activity of PLpro could therefore boost glycolytic flux, and promote the glycolysis-driven inflammatory phenotype of M1 macrophages. Interestingly, ISGylation has also been linked to metabolic control in Listeria infection [94]. 

Another important enzyme in the ISGylation machinery is USP18, which has similar functions to the viral protease PLpro as it is primarily involved in deconjugating ISG15. USP18 is the only described protease with deISGylating activity in vivo, and is well known as a potent negative regulator of IFN signaling [95]. Therefore, downregulating USP18 can have therapeutic benefits, particularly in the case of SARS-CoV-2, to counteract a decrease in IFN signaling and improve viral sensing and clearance. Interestingly, it remains unstudied if SARS-CoV-2 modulates USP18, as both the cell and the virus fight for the same ISGylated substrates.

## 4. Therapeutics-the Ub and ISG15 Machinery as a Target for COVID Treatment Regimes

Current pharmacological treatment regimens for COVID-19 can be broadly classified into antivirals, immunomodulators, and anticoagulant/antiplatelet therapies. Pharmacological regimens are primarily directed at mitigating complications in the later stages of the infection rather than early intervention during viral replication, and so far, only a handful have made it to clinical trials [96]. It is probable that many candidate compounds will have no therapeutic benefit in vivo; however, they can still provide invaluable insights into the druggability of the target, and act as molecular scaffolds in novel compound development. 

Viral proteases have proven to be an attractive class of druggable enzymes and inhibitors directed against them are being investigated as possible therapeutic agents that may specifically target pathways within the SARS-CoV-2 lifecycle, or modulate the host immune response. The following section will explore the therapeutic potency of some of the most promising host (human) and viral proteases within the ubiquitin and ISG15 pathways.

### 4.1. Host Proteases

The IFN-I response is one of the first protective measures taken to encourage viral clearance during infection. It acts on downstream signaling pathways that result in the transcription of ISGs, leading to interactions between the innate and the adaptive immune systems [97]. As mentioned previously, there is an abnormal IFN-I response when cells are infected by SARS-CoV-2 [98,99]. This sometimes delayed but powerful IFN-I response has severe consequences as it hinders viral clearance and could contribute to the paradoxical hyperinflammatory state associated with cytokine storming. This regulation is tissue specific. For instance, a study has found that IFNα induces ACE2 expression as an ISG selectively in primary human upper airway basal cells [100], highlighting the importance of the host regulation of the IFN-I pathway in SARS-CoV-2 infection. PLpro and host proteases such as OTULIN, OTUB1/2, A20, CYLD, USP4, USP15, and USP18 are likely to be involved in such responses as key regulators of the innate immune response. Inhibitors of host proteases involved in antiviral response have already been described and could be explored in the context of COVID-19 infection. For example, Subquinocin, a CYLD inhibitor, was able to enhance activation of NF-κB signaling in cells [101]. Small molecule inhibitors for OTUB2 (OTUB2-COV-1) [102] and USP4 (NR) [103] have also been identified. 

Small molecule inhibitors would be the traditional approach to downregulate the activity of an enzyme; however, as the case with some DUBs and most E3s, identifying suitable compounds that show high specificity whilst having limited off-target effects and good pharmacokinetic properties has been generally difficult [104,105]. Targeted degradation technologies such as degraders, and Proteolysis Targeting Chimeras (PROTACs) can be an alternative solution [106]. PROTACs enable a target protein to be ubiquitinated by an E3 ligase such that it undergoes proteasome-mediated degradation. This strategy has already been used for the FDA-approved drug ARV-110 which is used to treat metastatic castration-resistant prostate cancer [104]. The use of degraders or PROTACs could be the way to drug difficult-to-inhibit E3 ligases involved in antiviral responses such as TRIM25 and HERC5.

It may also be worth considering additional SARS-CoV-2-regulated host proteases, due to their likely involvement in inflammatory and immune response pathways. For instance, a study monitored lysine ubiquitylation on 8943 proteins from Vero E6 cells during SARS-CoV-2 infection by label-free LC-MS/MS [107]. Seventy-two hours post-infection, the authors observed a significant upregulation of 104 proteins and downregulation of a further 447. Several USP5 ubiquitin sites were downregulated during infection, whereas a single ubiquitylation (K558) site was upregulated. Similarly, in their global proteomic analysis, USP5 was upregulated by a factor of 1.2 [107]. This upregulation of USP5 provided insights into the deregulated IFN-I response during SARS-CoV-2 infection. The authors proposed a model whereby USP5 upregulation leads to the increased ubiquitylation of RIG-1, enabling the recruitment of STUB1 and inhibition of IFN-I production. Taken together, USP5 is an attractive target of pharmacological intervention that could be used in conjunction with other antivirals to induce a more adequate IFN response. Other targets of interest within the UPS include USP13, which has been reported to regulate IFN signaling via its ubiquitylation of STAT1 [108]. Furthermore, USP15 and USP25 are known negative regulators of IFN signaling [109,110]. Although most of the DUBs mentioned here have not been directly implicated in SARS-CoV-2 infection, they hold significant therapeutic potential.

### 4.2. Viral Proteases

SARS-CoV-2 encodes two proteases, the papain-like cysteine protease (PLpro) and the 3-chymotrypsin-like protease (3CLpro or Mpro) [111] and, unsurprisingly, both have garnered much attention in the exploration of therapeutic targets. 

#### 4.2.1. PLpro inhibitors

PLpro enables viral protein maturation through the cleavage of the viral polyprotein, and modulates host immune response via deubiquitylation and removal of ISG15 [72,85]. In an effort to find an effective pharmacological inhibitor against it, one study scanned a library of 3727 approved drugs and late-stage clinical drug candidates to repurpose, but failed to find any compounds which successfully inhibited the target [88]. Existing SARS PLpro inhibitors (rac3j, rac3k and rac5c) have been tested against SARS-CoV-2 PLpro. The most effective was rac5c, which was also shown to have acceptable cell toxicity, and decreased viral titers to similar levels as remdesivir and hydroxychloroquine regimes [88]. GRL0617 is a SARS PLpro inhibitor and has also been tested against SARS-COV-2 PLpro [91]. The study obtained an IC50 of 2.1 ± 0.2 μM for the compound, and also demonstrated that GRL0617 was capable of inhibiting the deubiquitylation and deISGylation activity of PLpro in HEK293T cells. GRL0617 is highly selective and does not alter the activity of USP18, the host-encoded deISGylating enzyme [91,112]. Furthermore, this drug was shown to have limited cytopathic effect, with good inhibition of viral replication in Vero E6 cells [91,113]. 

Molecular modelling approaches are also being used to find PLpro inhibitors. One group investigated 50 compounds with at least 65% similarity to GRL0617 through a molecular docking simulation [114]. They identified four potential inhibitors, which also had acceptable water solubility, toxicity, and gastrointestinal absorption [114]. Another study used similar methods to monitor the inhibitory potential of 67 approved compounds against PLpro. Of these, 26 had superior inhibitory potential compared with the GRL0617control [115]. Improving on the initial results, this latter study used the actual structure of PLpro rather than relying on homology modelling. It must be noted, however, that these drugs have not yet been approved for use in clinical trials. 

Natural biflavones have also been reported as potential inhibitors of SARS-CoV-2 PLpro. Various phytochemicals such as polyphenols derived from plants such as *Broussonetia papyrifera* and *Paulownia tomentosa* have been shown to affect SARS-CoV-1 and MERS-CoV proteases [116,117]. Nine naturally occurring biflavones were identified from the national library of traditional Chinese medicine [118], and selected based on their inhibitory potential in molecular docking simulation against SARS-CoV-2 PLpro and subsequent validation by fluorogenic enzymatic assays. Of these, 4′-O-metyhlochnaflavone showed 60.7% inhibition even at concentrations as low as 2.5 μM, which is noteworthy as it is derived from *L. japonica*—a plant that has already shown to have antiviral effects against CoVs [118,119]. Other natural compounds such as ginkgolic acid and anacardic acid have been demonstrated to inhibit PLpro and have antiviral activity in vitro, with ginkgolic acid exhibiting higher antiviral potency amongst the two [120]. Tannic acid and methylcobalamin have also shown potent inhibitory effects on PLpro, independent of their other antiviral mechanisms [120,121,122].

Nanobodies (NbSL-17, -18 and -19) have been developed that targeted the S1 pocket of PLpro, a site that contributes to its deubiquitinating and deISGylating activity. NbSL-18 was shown to inhibit the cleavage of K48-linked triUB in a dose-dependent fashion, although there was less inhibition of ISG15-cleaving activity [123]. Interestingly, NbSL-18 inhibits Nsp3 mediated viral polyprotein processing, similar to rac5c [88,123]. It has been suggested that additional binding sites on PLpro distal from the active-site cysteine would allow development of compounds that exploit this binding cooperativity [124]. This included development of XR8-24, which could inhibit SARS-CoV-2 replication in human alveolar basal epithelial A549 cells. There was no inhibition of USP7 (the most similar human DUB to PLpro in structural terms), suggesting high target specificity. In addition, the authors argue that slower ligand dissociation and slow-off rates could be a possible consequence of positive binding cooperativity [124].

An attempt was made to repurpose 1971 FDA-approved drugs [123]; however, despite having identified five potential inhibitory compounds against SARS-CoV-2 PLpro and nsp3, these compounds demonstrated considerable off-target effects and low antiviral potency in cell-based models. More successful efforts have been made to repurpose approved, safe, cysteine modifiers that can inhibit the catalytic cysteines of PLpro, such as Disulfiran and Ebselen, two molecules that are currently part of the COVID-19 clinical trials (NCT04485130, NCT04483973 and NCT04484025) [125]. Another potential PLpro inhibitor undergoing clinical trials is Isotretinoin, a retinoic acid used for the treatment of severe acne (NCT04389580, NCT04382950, NCT04353180 and NCT04361422). This is encouraging and further supports targeting PLpro for the treatment of COVID-19.

#### 4.2.2. CLpro/Mpro Inhibitors 

Mpro is responsible for cleaving the majority of the viral polyprotein. Although it holds promise as a therapeutic target, we will only briefly summarize these findings as it does not play a known role in modulating host ubiquitin/ISG15 response. Different MPro inhibitors have been characterized [115,120,124,126,127,128,129,130,131,132,133,134]); however, the most successful has been Paxlovid, developed by Pfizer (PF-07321332) [126]. Paxlovid is an orally bioavailable Mpro inhibitor with in vitro and in vivo antiviral activities that has been tested in patients in combination with ritonavir (a protease inhibitor that extends the half-life of PF-07321332 in the body). As the results of a recent Phase 2/3 clinical trial show, paxlovid has been found to reduce the risk of hospitalization or death by 89% compared with placebo in non-hospitalized high-risk adults with COVID-19. Paxlovid has just been approved by the U.S. Food and Drug Administration issued with an emergency use authorization (EUA) and by the Medicines and Healthcare products Regulatory Agency (MHRA) in the UK (https://www.fda.gov/news-events/press-announcements/coronavirus-COVID-19-update-fda-authorizes-first-oral-antiviral-treatment-COVID-19; 28 January 2022). PF-07321332 is based on a previous compound (PF-835231) that was developed by Pfizer against SARS in 2003/4. Despite being based on an existing molecule, it is impressive that in only a matter of months, PF-07321332 was designed, synthesized, tested in vitro and in vivo, and subsequently approved for use in the clinic.

## 5. Conclusions and Future Perspectives

SARS-CoV-2 is a positive-sense single-stranded RNA virus, and it is the causative virus of COVID-19 in humans. Owing to increasing numbers of fatalities, the pandemic necessitates urgent therapeutic intervention. Genomic and proteomics analyses have been key in the design and implementation of specific and safe SARS-CoV-2 vaccines and other treatments such as monoclonal antibodies that have saved numerous lives (https://www.COVID19treatmentguidelines.nih.gov/therapies/anti-sars-cov-2-antibody-products; 28 January 2022). The severity of COVID-19 can widely vary between patients. Innate immune sensing and the subsequent activation of the adaptive immune response seem to play a critical role in this [31].

Machinery of the innate immune response is tightly regulated by selective expression of innate immune factors and PTMs of key enzymes in host tissues. For efficient infection, viruses need to evade the innate immune response. For this, they very often hijack the conjugation/deconjugation of ubiquitin and ISG15 on key viral and host proteins. Many virus proteases are able to revert such modifications encoding for Ub/ISG15 proteases. SARS-CoV-2 modulates and inhibits the host ubiquitin and ISG15 response through the expression of PLpro, which cleaves both, ubiquitin and ISG15 from proteins, disrupting antiviral functions. In addition, a number of host factors inside the ubiquitin system machinery interact with SARS-CoV-2 proteins [82,135] and several viral proteins are ubiquitylated [136]. Thus, it appears that SARS-CoV-2 attempts to take control of the host ubiquitin and ISG15 systems.

Viral proteases have been successfully used as targets for small-molecule therapies for the treatment of HIV and HCV [137,138]. The success of paxlovid supports the potential of using viral proteases in COVID-19. The ability of PLpro to inhibit immune responses mediated by ISG15 makes it a promising target for COVID-19 therapeutics. A number of PLpro inhibitors such as GRL0617 have shown antiviral efficacy against SARS-CoV-2 in vitro; however, it is too soon to know if they have the same effects in pre-clinical studies. Approved drugs with described and potential PLpro inhibition profiles as mechanism of action such Ebselen, Disulfiran and Isotretinoin are currently part of COVID-19 clinical trials. These drugs, despite not being highly selective against PLpro, suggest that targeting the deubiquitylating/deISGylating activities of PLpro can be a valid strategy for the treatment of COVID-19.

In addition to targeting viral proteins, it is worth considering host enzymes modulating ubiquitylation/ISGylation of innate immune regulators as targets to neutralize SARS-CoV-2 hijacking of cell functions. Not only can they prevent undesirable host–pathogen interactions, but as host proteins evolve at a much slower pace than ever-mutating viral proteins, they also provide a more stable form of therapy. For instance, USP18 is a potentially relevant target. Similar to PLpro, USP18 also cleaves ISG15 from proteins, acts as a key negative regulator of the interferon pathway, and has been linked to resistance to pathogens [104]. Therefore, USP18 could help to increase antiviral responses and to modulate cytokine storms. It would be also interesting to study whether there is a modulation of USP18 activity during COVID-19 infection as both PLpro and USP18 proteins share substrates. This could provide new avenues of therapeutic intervention. Interestingly, different ways could be used to target USP18: via catalytic activity inhibition with small molecules to prevent deISGylation; by blocking the inhibitory interaction with STAT2 with peptides [139]; or by inducing degradation of USP18, thus preventing both activities, either via degraders/PROTACs or by preventing stabilization by interaction with ISG15 [140]. 

Deubiquitylating enzymes are druggable enzymes and are currently being explored as pharmacological targets for other diseases [141,142]. Together with USP18, other enzymes with important roles in innate immune responses such as USP13, USP15, USP25, OTUB1, OTUB2, OTULIN, A20, and CYLD represent attractive antiviral targets, and inhibitors for some of them are already available (highlighted in Figure 2). Other DUBs, such as USP8, have been identified as interactors of the viral ORF3 protein and could have a role in viral infection [82]. In addition to DUBs, E3 ligases are equally interesting therapeutic targets. E3 ligases initiate ubiquitin and Ubl signaling during SARS-CoV-2 and other viral infections. Despite their central importance, E3 ligases remain challenging therapeutic targets as their catalytic activity is not easily inhibited by small molecules. However, small molecules promoting the protein degradation (PROTACs) of E3 ligases have already been used in the clinics and could be developed against SARS-CoV-2-relevant E3 enzymes.

It is important to mention that targeting components of the innate immune response could also lead to undesirable side effects, and this should always be considered in studies of a preclinical nature. For example, inherited inactivating mutations of USP18 have been linked to type I interferonopathies [143,144]. Immune-related toxicities are observed when targeting immunomodulatory proteins and can be severe. In the specific case of immune checkpoint inhibitors for cancer immunotherapy, these have been termed immune-related adverse events (irAEs) [145].

In summary, ubiquitylation and ISGylation are two important mechanisms underlying host–pathogen interactions and the defense response of humans infected with viruses such as SARS-CoV-2. The virus hijacks these pathways by either removing ubiquitin and/or ISG15 from proteins, but also by interacting with enzymes involved in ubiquitin/ISG15 machinery. We have highlighted likely points of therapeutic intervention for the treatment of COVID-19 that may prove helpful to combat future vaccine-resistant variants of SARS-CoV-2.

## Figures and Tables

**Figure 1 biomolecules-12-00300-f001:**
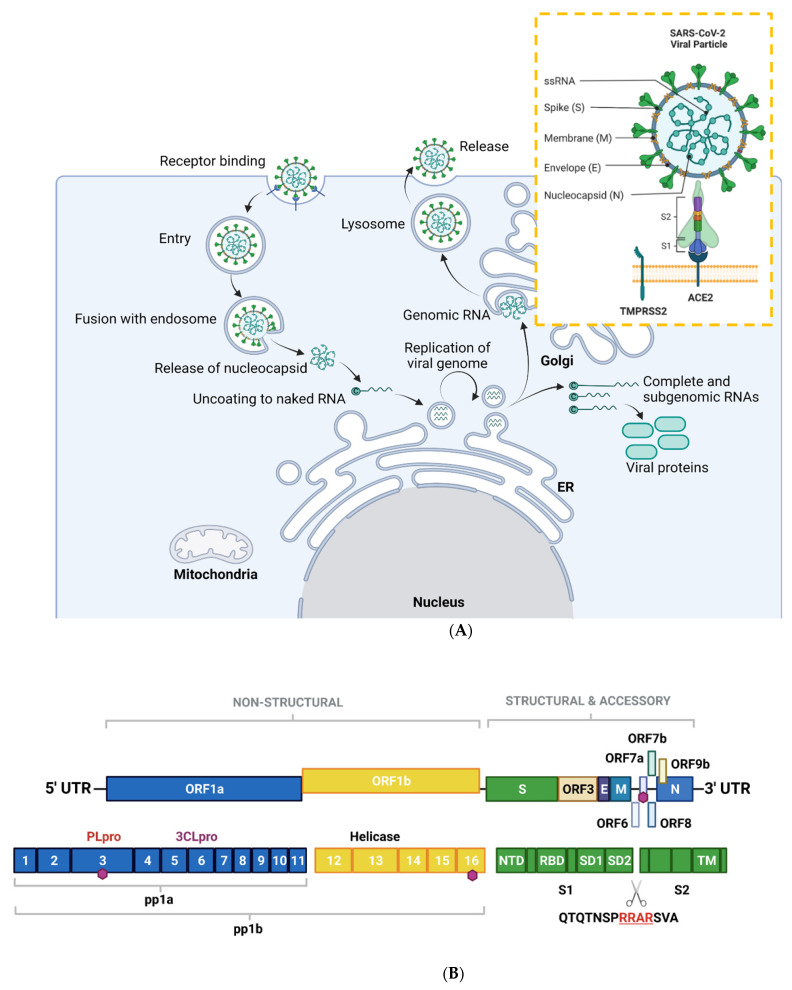
**Overview of the SARS-CoV-2 viral life cycle and structural organization**. (**A**) SARS-CoV-2 comprises four structural proteins—the envelope (E) and membrane (M) proteins encase the single-stranded RNA and nucleocapsid (N). The spike (S) protein is made up of S1 and S2 subunits and facilitates receptor binding in the host cell. In humans, this is ACE2, in synergy with TMPRSS2 (orange box). SARS-CoV-2 binds to ACE2 on respiratory epithelial cells through the receptor binding domain on S. The SARS-CoV-2 viral particle is endocytosed following cleavage of S. SARS-CoV-2 nucleocapsid, a complex of viral RNA and N protein, is released into the cytosol. The N protein is removed from the capsid, leaving naked viral RNA, which is a capped positive-stranded RNA molecule that can immediately be translated to allow production of viral proteins (not shown). Viral replication centers are formed in the ER, using the positive-stranded RNA molecule as a template to form a double-stranded RNA intermediate, which in turns allows replication of the viral genome. Complete and sub-genomic RNAs are also formed during the replication process, which serve as the basis for production of a range of viral proteins. The replicated genomic RNA is then bound by N protein to form nucleocapsid, which is encapsulated in vesicles at the Golgi. The contents of the vesicles are then exocytosed, spreading viral particles from the infected cell. (**B**) Layout of SARS-CoV-2 genome. The virus encodes for ORF1a, ORF1b, ORF3, ORF6, ORF7a/b, ORF8, ORF9b, as well as S, E, M, N. Sixteen non-structural proteins (nsps1–16) of varying function are encoded by OF1a and ORF1b. nsp1 regulates viral mRNAs and interferes with host translation, nsp2–11 facilitate viral replication, nsp12 has RNA polymerase activity, whereas nsp14 is involved in RNA proofreading. A scissors indicates the site of furin and TMPRSS2 cleavage in S1/S2 and S2′, whereas sites of ubiquitin modification are identified by purple hexagons. Other regions of interest include PLpro and 3CLpro. S is composed of several sub-domains including the N-terminal domain (NTD), receptor-binding domain (RBD), subdomains 1 and 2 (SD1, SD2) and the transmembrane domain (TM). Panel B is adapted from Zhang et al., 2021. Figure created with BioRender.com in January 2022.

**Figure 2 biomolecules-12-00300-f002:**
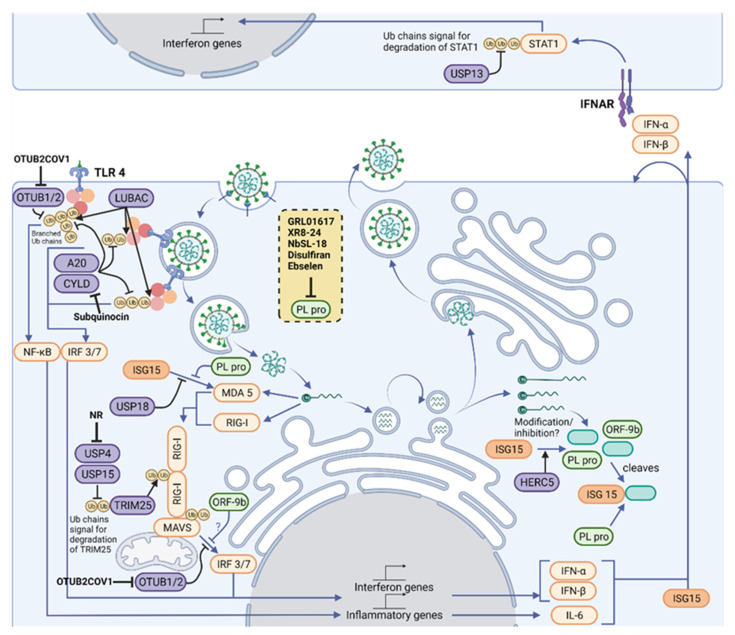
**Therapeutic intervention points in the Ubiquitylation and ISGylation machinery of the host and of SARS-CoV-2**. Polyubiquitin chains are vital mediators in signaling downstream of three of the main families of innate immune signaling pathways, TLRs, RLRs and NLRs, which act together to sense a variety of PAMPs, including SARS-CoV-2. A network of E3 ligases and DUBs (purple) generate new ubiquitin (yellow) and ISG15 (orange) architectures during viral infection. For instance, LUBAC assembles linear ubiquitin chains on pre-existing K63-linked ubiquitin chains, forming branched structures. Their structures are reshaped and limited by OTUB1/2, OTULIN (not shown), A20 and CYLD. K63-linked ubiquitin chains are assembled on RIG-I by TRIM25, which allow its polymerization and activation of MAVS. In addition, the amount of K48-linked ubiquitin chains regulates TRIM25 levels, and these chains are cleaved by USP4 and USP15. MAVS signaling to IRF3/7 depends on ubiquitylation of NEMO (not shown), but this can be removed by OTUB1/2. Some of these pathways end up activating the IFN-I pathway, driving production of ISGs, including more IFN and ISG15. ISG15 is conjugated by HERC5 to host proteins, which activates antiviral signaling, and to viral proteins, which is believed to inhibit viral processes. Signaling from MDA5 and RIG-I to MAVS depends on the ubiquitylation of MAVS and RIG-I, as well as on the ISGylation of MDA5. Activated MAVS drives IRF3/IRF7-dependent IFN-I gene production. However, the cell also produces USP18, an ISG protease that removes ISG15 from cellular proteins, fine-tuning IFN signals. An antiviral response is driven by neighboring cells through paracrine signaling of type I IFNs. The extent of downstream signaling depends on levels of STAT1, which is modulated by K48-linked ubiquitin chains and its removal by USP13. Host protease inhibitors have been described for CYLD (Subquinocin), OTUB2 (OTUB2COV1) and USP4 (NR). SARS-CoV-2 opposes ubiquitin- and ISG15-mediated antiviral mechanisms by encoding the multi-activity deubiquitylating/deISGylating enzyme PLpro (light green) to inhibit their function. A number of PLpro inhibitors have been identified (listed in the yellow box), characterized and are undergoing clinical trials. Figure created with BioRender.com in January 2022.

## Data Availability

Not applicable.

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
