# Peer review of "Targeting the Ubiquitylation and ISGylation Machinery for the Treatment of COVID-19"

_biomolecules, 2022, doi:10.3390/biom12020300_

Round 1

Reviewer 1 Report

This is an excellent review which comprehensively and in great detail describes the current knowledge of  of ubiquitin and ISG15 modification within the context of SARS-CoV 2infection. Background information about molecular mechanisms and therapeutic options are clearly depicted and it is real fun to read this paper.

I have only some minor suggestions which might be added to this extremely  well written manuscript :

line 257/8 as authors previously discussed differences in mouse and human ISG15, they might also mention here that the major E3s for ISGylation differs between mouse and humans (HERC5 vs HERC6)

line 315 Add references for the structural analysis

line 417 Typo: "of"  missing

line 524 (Discussion) Here it might be interesting to note that USP18  can be adressed therapeutically either to inhibit deISGylation activity (e.g by blocking the catalytic activity,) adress the influence on the IFN response (e.g by blocking STAT2 interaction or target both activities (e.g by promoting the degradation of USP18 via PROTACS or preventing interaction with free ISG15 that is essential to stabilize hUSP18)

Author Response

We would like to thank the referee for expressing a very positive opinion about our work and for providing constructive feedback during the review process.

As for your specific comments:

line 257/8 as authors previously discussed differences in mouse and human ISG15, they might also mention here that the major E3s for ISGylation differs between mouse and humans (HERC5 vs HERC6)

We thank the reviewer for this suggestion and we have now updated the text to include it (lines 261-263)

line 315 Add references for the structural analysis

Apologies for this error, we have now added two references to support the text (lines 317-318)

line 417 Typo: "of"  missing

Apologies for this error, this has been now corrected (line 424)

line 524 (Discussion) Here it might be interesting to note that USP18  can be adressed therapeutically either to inhibit deISGylation activity (e.g by blocking the catalytic activity,) adress the influence on the IFN response (e.g by blocking STAT2 interaction or target both activities (e.g by promoting the degradation of USP18 via PROTACS or preventing interaction with free ISG15 that is essential to stabilize hUSP18)

We thank the reviewer for raising this interesting point. We have now discussed this in the text (lines 533-538)

Reviewer 2 Report

The manuscript of George Vere and co-authors is relevant and interesting. The authors describe the mechanisms of infection with SARS-CoV-2, as well as mechanisms of Ubiquitylation, ISGylation and roles of these processes in human antiviral responses and viral modulation of ubiquitin and ISG15 signals. The text is well illustrated. Based on the described mechanisms, the authors believe that the Ub and ISG15 machinery may be a target for COVID treatment. The manuscript may be useful for investigators in this field.

However, I have some critical notes.

- In the abstract, the authors briefly state: “In this review, we highlight potential points of therapeutic intervention in ubiquitin/ISG15 pathways that could be targeted in the treatment of COVID-19 infection and which may prove effective in combatting current and future vaccine-resistant variants of the disease”. I think, in the abstract it should be specified which proteins involved in these two pathways may be the most promising as targets. In this case, the abstract will correspond more to the manuscript title.

- In chapter “Conclusions and Future Perspectives”, the authors should specify possible limitations of the proposed targeting.

- The authors use in the manuscript two versions of the process name: “ubiquitylation” and “ubiquitination”. It would be better to bring the spelling to uniformity.

Author Response

We would like to thank the referee for expressing a very positive opinion about our work and for providing constructive feedback during the review process.

As for your specific comments:

- In the abstract, the authors briefly state: “In this review, we highlight potential points of therapeutic intervention in ubiquitin/ISG15 pathways that could be targeted in the treatment of COVID-19 infection and which may prove effective in combatting current and future vaccine-resistant variants of the disease”. I think, in the abstract it should be specified which proteins involved in these two pathways may be the most promising as targets. In this case, the abstract will correspond more to the manuscript title.

We thank the reviewer for this suggestion and we have now included in the abstract some examples of the targets we discuss in the main text (line 23)

- In chapter “Conclusions and Future Perspectives”, the authors should specify possible limitations of the proposed targeting.

This is a very important point and we thank the reviewer for his comment. We have now discussed limitations in terms of toxicity and side effects of the suggested targets (lines 552 -558)

- The authors use in the manuscript two versions of the process name: “ubiquitylation” and “ubiquitination”. It would be better to bring the spelling to uniformity.

Apologies for this inconsistency. This has been now amended across the text.